# A Metabolically Healthy Profile Is a Transient Stage When Exercise and Diet Are Not Supervised: Long-Term Effects in the EXERDIET-HTA Study

**DOI:** 10.3390/ijerph17082830

**Published:** 2020-04-20

**Authors:** Pablo Corres, Simon M. Fryer, Aitor Martínez Aguirre-Betolaza, Ilargi Gorostegi-Anduaga, Iñaki Arratibel-Imaz, Javier Pérez-Asenjo, Silvia Francisco-Terreros, Ramón Saracho, Sara Maldonado-Martín

**Affiliations:** 1Department of Physical Education and Sport, Faculty of Education and Sport-Physical Activity and Sport Sciences Section, University of the Basque Country (UPV/EHU), 01007 Vitoria-Gasteiz, Araba/Álava, Basque Country, Spain; pablo.corres@ehu.eus (P.C.); aitor.martinezdeaguirre@ehu.eus (A.M.A.-B.); ilargi.gorostegi@ehu.eus (I.G.-A.); inaki.arratibel@ehu.eus (I.A.-I.); 2School of Sport and Exercise, Oxstalls Campus, University of Gloucestershire, Gloucester GL2 9HW, UK; sfryer@glos.ac.uk; 3Cardiology Unit, Igualatorio Médico Quirúrgico (IMQ-Amárica), 01005 Vitoria-Gasteiz, Araba/Álava, Basque Country, Spain; jpasenjo@ubikare.io; 4Clinical Trials Unit, Health and Quality of Life Area, TECNALIA, 01009 Vitoria-Gasteiz, Araba/Álava, Basque Country, Spain; SILVIA.FRANCISCOTERREROS@osakidetza.eus; 5Nefrology Department, Osakidetza, Hospital University of Araba, 01009 Vitoria-Gasteiz, Araba/Álava, Basque Country, Spain; rmsaracho@gmail.com

**Keywords:** obesity, hypertension, inactivity, supervised exercise

## Abstract

Metabolically unhealthy obesity (MUO) is a regular state in people with primary hypertension (HTN), obesity, and who are physically inactive. To achieve and maintain a metabolically healthy overweight/obese (MHO) state should be a main treatment goal. The aims of the study were (1) to determine differences in metabolic profiles of overweight/obese, physically inactive individuals with HTN following a 16-week (POST) supervised aerobic exercise training (SupExT) intervention with an attentional control (AC) group, and (2) to determine whether the changes observed were maintained following six months (6 M) of unsupervised time. Participants (n = 219) were randomly assigned into AC or SupExT groups. All participants underwent a hypocaloric diet. At POST, all participants received diet and physical activity advice for the following 6 M, with no supervision. All measurements were assessed pre-intervention (PRE), POST, and after 6 M. From PRE to POST, MUO participants became MHO with improved (*p* < 0.05) total cholesterol (TC, ∆ = −12.1 mg/dL), alanine aminotransferase (∆ = −8.3 U/L), glucose (∆ = −5.5 mg/dL), C-reactive protein (∆ = −1.4 mg/dL), systolic blood pressure (SBP), and cardiorespiratory fitness (CRF) compared to unhealthy optimal cut-off values. However, after 6 M, TC, glucose, and SBP returned to unhealthy values (*p* < 0.05). In a non-physically active population with obesity and HTN, a 16-week SupExT and diet intervention significantly improves cardiometabolic profile from MUO to MHO. However, after 6 M of no supervision, participants returned to MUO. The findings of this study highlight the need for regular, systematic, and supervised diet and exercise programs to avoid subsequent declines in cardiometabolic health.

## 1. Introduction

Obesity is a complex and chronic non-communicable disease with a disparity in the way it is classified [1]. The combination of obesity, physical inactivity, and primary hypertension (HTN) is widely recognized as a pre-eminent cause of cardiovascular risk and metabolic complications [2,3], and is termed ‘metabolically unhealthy obesity’ (MUO) [4,5]. Therefore, to become metabolically healthy overweight/obese (MHO) (i.e., overweight/obesity in the absence of a clearly defined cardiometabolic disorder and/or high level of cardiorespiratory fitness (CRF)) [6,7] should be one of the main priorities in the treatment of this population. A previous systematic review and meta-analysis found that MHO individuals had significantly higher levels of physical activity and CRF, and spent less time in sedentary behaviour than MUO, suggesting that a healthier metabolic profile could be partially due to a healthier lifestyle [5].

Major international guidelines recommend non-pharmacological, tailored, and long-term lifestyle changes for the prevention and treatment of HTN and obesity in this population [8,9,10]. Previously, interventions with engagement in regular physical activity, exercise, and a healthy diet in overweight/obese individuals with HTN reported significantly reduced blood pressure (BP) [10,11], and improved body composition [8,11], CRF [11,12], and biochemical profile [12,13]. In addition, it was suggested that physical activity and diet recommendations were not enough to improve biochemical profile alone and that supervision was needed [14]. As such, education programs (including healthy diet and physical exercise) should be regular, systematic, and supervised by specialists. This is particularly important since previously it has been found that declines in cardiometabolic health occur after finishing a time-limited exercise program in different populations [15,16,17].

Previous data from the EXERDIET-HTA investigation found that all study groups improved body composition, BP, and CRF following a 16-week intervention program [11]. However, there was a significant reduction in the improvements found after the 6-month (6 M) follow up, when the exercise and diet supervision were removed and only recommendations were applied [14]. Nevertheless, no known study has analysed the changes in metabolic profile in an overweight/obese population with HTN. Accordingly, the aims of the study were (1) to determine differences in metabolic profiles of overweight/obese, physically inactive individuals with HTN following a 16-week supervised aerobic exercise training (SupExT) intervention with an attentional control (AC) group, both with a hypocaloric diet, and (2) to analyse whether the changes observed during supervision were maintained following 6 M of unsupervised time. 

## 2. Methods

### 2.1. Research Design

The EXERDIET-HTA study is a randomised controlled trial (ClinicalTrials.gov ID: NCT02283047) that compares the immediate (POST) and 6 M-effects of different 16-week SupExT programmes (performed 2 days/week) combined with a dietary intervention in physically inactive, overweight/obese individuals with HTN. The Clinical Investigation of Araba University Hospital (2015-030) and the Ethics Committee of The University of the Basque Country (UPV/EHU, CEISH/279/2014) approved the study design, protocols, and informed consent. More details regarding the design, selection criteria, and procedures of the EXERDIET-HTA study have been explained in previous publications [11,14,18].

### 2.2. Participants

The EXERDIET-HTA study was conducted in Vitoria-Gasteiz (Basque Country, Spain). Non-Hispanic white adult participants (*n* = 219) took part in the study, but 23 participants left the study during the intervention and 19 participants did not attend the 6 M follow-up visit. These participants did not differ in any way from the main sample. As such, 177 participants (n = 114 men and n = 63 women, 51.6 ± 8.9 years) were included in the analysis. Figure 1 represents the participants and design of the EXERDIET-HTA study from recruitment to 6 M post intervention. The main inclusion criteria were being physically inactive and having overweight/obesity with primary HTN. The International Physical Activity Questionnaire (IPAQ) determined physical activity behaviour [19], and participants were below the “*Global Recommendations on Physical Activity for Health*” set by the World Health Organization [20]. Body mass index (BMI) had to be above 25 kg∙m^2^ [8]. Primary HTN was defined as systolic BP (SBP) of 140–179 mmHg and/or diastolic BP (DBP) of 90–109 mmHg, and/or under antihypertensive pharmacological treatment [21]. For all other inclusion and exclusion criteria, please refer to the previously published study protocol [18].

In addition to the EXERDIET-HTA study participants, another group was created to allow for comparison to a normal body mass healthy population (hereinafter termed: HEALTHY). The HEALTHY (n = 31, 40.0 ± 9.0 years, 58% women) group did not participate in any intervention. Only baseline measures were assessed for comparison to the EXERDIET-HTA study participants. Inclusion for HEALTHY criteria were age (25–55 years) and exclusion criteria were being pregnant, currently breastfeeding, taking regular medication, or having any known medical condition. 

### 2.3. Measurements

Anthropometric, clinical, and physiological measurements were taken at baseline (PRE), immediately after the 16-week intervention (POST), and at 6 M follow-up (Figure 1) by trained investigators and specialists. Each assessment included anthropometry (body mass, stature, BMI, waist circumference, and body composition), 24 h ambulatory BP monitoring, determination of peak oxygen uptake ($\overset{\cdot }{V}$O_2peak_) to asses CRF, and collection of fasting blood samples (12.5 mL) at the Clinical Trials Unit of Tecnalia (HUA, Vitoria-Gasteiz). The fasting blood samples were used to determine the metabolic profile which consisted of: C-reactive protein (CRP), aspartate aminotransferase (AST), alanine aminotransferase (ALT), gamma-glutamyl transpeptidase (GGT), total cholesterol (TC), high-density lipoprotein cholesterol (HDL-C), low-density lipoprotein cholesterol (LDL-C), triglycerides (TG), glucose, insulin, and haemoglobin A1c (HbA1c). HOmeostatic Model Assessment-Insulin resistance index (HOMA-IR) was determined by: fasting serum insulin (µU/mL) × fasting plasma glucose (mg/dL)/405 [22]. For more details of the assessments, please refer to the study protocol [18] and previous cardiometabolic profiling manuscript [14].

The cut-off points of parameters related to cardiometabolic abnormalities were: concentrations of CRP > 3 mg/L [23]. Hepatic enzymes when AST > 30 U/L, ALT > 30 U/L, GGT > 50 U/L [24]. With respect to the lipid profile, when TC > 200 mg/dL (52 mmol/L), LDL-C > 100 mg/dL (26 mmol/L), HDL-C < 40 mg/dL (10.4 mmol/L), TG > 200 mg/dL (2.28 mmol/L), and TC/HDL-C ratio > 3.5 [25]. Based on the Diabetes Federation Statement [26], glucose > 100 mg/dL (5.55 mmol/L). The HOMA-IR ratio cut point was established at 3.8, insulin cut point at 16.7 mU/L, and HbA1c at 6% [27,28]. A CRF ($\overset{\cdot }{V}$O_2max_ in mL·kg^−1^·min^−1^) reference value lower than the 50th percentile was used as the cut-off point, according to the FRIEND Registry [29]. The definition of the MUO and the MHO phenotypes were obtained based on the joint combination of obesity markers (i.e., BMI) and cardiometabolic abnormalities, taking into account the following definitions: the National Cholesterol Education Program-Adults Treatment Panel III, Wildman, Wildman Modified and Ortega [4,5,30]. 

### 2.4. Intervention and 6-Month Post-Intervention Follow-up

After baseline measurements, participants were randomly allocated into one of the intervention groups stratified by sex, SBP, BMI, and age using a time-blocked computerised randomisation program. All participants underwent a hypocaloric DASH diet (Dietary Approaches to Stop Hypertension) [13]. Habitual food consumption and nutrient intake were evaluated using three questionnaires: Dietary History, Food Frequency Questionnaire, and 24 h Recall Questionnaire. Every 2 weeks, participants were weighed and received encouragement and advice alongside nutritional counselling in order to aid compliance. The four intervention groups were AC group (AC, participants were given physical activity advice only), and three SupExT groups: high-volume and moderate-intensity continuous training, high-volume and high-intensity interval training, and low-volume and high-intensity interval training. These groups trained on 2 non-consecutive days under supervision by exercise specialists for 16-weeks. The advice for the AC group was to participate in at least 30 min of moderate-intensity aerobic exercise (walking, jogging, cycling, or swimming) for 5–7 days per week blended with some dynamic resistance exercises [18].

In a preliminary analysis of the data used in this article, SupExT groups had no significant differences (*p* > 0.05) among them in the target variables (biochemical profile variables) over time (i.e., PRE versus POST, POST versus 6 M, and PRE versus 6 M). Therefore, for the purposes of this article, the three SupExT groups were put together in one group and thus comparative analyses were performed between groups (AC versus SupExT).

After POST assessment, all participants received physical activity and diet advice for the following 6 M. Participants had no further supervised intervention or attention from any of the research staff. All participants received exercise intensity domains (i.e., individual heart rate values at moderate- and high-intensity ranges) to self-monitor. All the protocols for each group, including procedures and diet intervention, have been previously published [11,18].

### 2.5. Statistical Analysis

Descriptive statistics were calculated for all variables and presented as mean ± standard deviation (SD) or percentage. To determine normality, a Kolmogorov–Smirnov test was performed on all variables, and those with a skewed distribution were log-transformed prior to any analysis. For comparisons between HEALTHY and the EXERDIET-HTA study population, independent *t*-tests were used to assess mean differences for continuous variables, and the chi-square test was performed to verify frequency differences for categorical variables. One-way repeated measures analysis of variance (ANOVA) was used to test the change in biochemical profile variables over time, when the differences presented significance, the Bonferroni post hoc test was applied. From the repeated measures ANOVA, partial eta-squared (η_p_^2^) was reported as a measure of effect size (ES). Two-way repeated measures ANOVA was used to evaluate the interaction effects (time × group) in biochemical profile variables. Analysis of covariance (ANCOVA) was performed to test the differences between groups for the delta score (Δ, differences between PRE versus POST, POST versus 6 M follow-up, and PRE versus 6 M follow-up), adjusting the analysis for age, sex, medication intake, and changes in body mass. Cochran’s Q test was executed to analyse the change in medication intake and smoking status. Chi-square test was used to test the differences between groups for the change in medication intake and smoking status (differences between PRE versus POST, POST versus 6 M follow-up, and PRE versus 6 M follow-up) over time. As previously described in the study protocol [18], the required sample size was determined for the primary outcome variable (SBP) of the EXERDIET-HTA study. It was identified that adequate power (0.95) to evaluate differences in our design consisting of four experimental groups would be achieved with 164 people (41 each group, α = 0.05, ES f = 0.4). Data were analysed according to the intention-to-treat principle. All statistical analyses were performed using the Statistical Package for Social Science (SPSS) version 24.0 (IBM Corp., Armonk, NY, USA). For all analysis, the alpha level of significance was set at *p* < 0.05.

## 3. Results

Table 1 presents baseline anthropometric, BP, CRF, biochemical profile, medication intake, and smoking status data of the EXERDIET-HTA study population compared to the HEALTHY group. The EXERDIET-HTA study population had significantly higher (*p* < 0.05) age, BM, BMI, waist circumference, SBP, DBP, mean BP (MBP), CRP, AST, ALT, GGT, TC, LDL-C, TG, TC/HDL-C, glucose, insulin, HOMA-IR, and HbA1C compared to the HEALTHY group. Nevertheless, the EXERDIET-HTA group had significantly lower fat-free mass, $\overset{\cdot }{V}$O_2peak_, and HDL-C than the HEALTHY group. There was no difference in smoking status between groups. As previously described [31], the participants from the EXERDIET-HTA study were considered MUO with the following parameters showing concentrations and values with cut-off points outside of those considered healthy: LDL-C, TC, TC/HDL-C, ALT, glucose, BP, CRP, and low CRF.

### 3.1. PRE versus POST Changes

Table 2 presents metabolic profile data at PRE, POST, and after 6 M follow-up. Analysing the change in group from PRE to POST, a one-way repeated measures ANOVA found that the SupExT group reduced concentrations of CRP (mean difference, Δ = −1.5 mg/L, 95% confidence interval (CI) = −2.4, −0.5 mg/L), ALT (Δ = −9.1 U/L, 95% CI = −14.3, −3.8 U/L), GGT (Δ = −9.1 U/L, 95% CI = −12.1, −5.4 U/L), TC (Δ = −13.9 mg/dL, 95% CI = −20.4, −7.4 mg/dL), LDL-C (Δ = −9.5 mg/dL, 95% CI = −15.5, −3.4 mg/dL), TG (Δ = −20.9 mg/dL, 95% CI = −30.3, −11.6 Mg/dL), TC/HDL-C (Δ = −0.3, 95% CI = −0.5, −0.1), glucose (Δ = −6.7 mg/dL, 95% CI = −12.0, −1.5 mg/dL), insulin (Δ = −2.2 mU/L, 95% CI = −3.9, −0.4 mU/L), HOMA-IR (Δ = −0.9, 95% CI = −1.5, −0.3), and HbA1C (Δ = −0.2%, 95% CI = −0.4, −0.1%). The AC group had reduced concentrations of CRP (Δ = −1.0, 95% CI = −1.8, −0.1), while there was no change in any other biochemical variable. Although the analysis by group found that the SupExT group had a greater reduction than the AC group, there was no interaction of time by group for PRE versus POST. Considering biochemical profile at POST, and the results previously presented for BMI, BP, and CRF [11], at POST, participants in the present study were considered MHO (i.e., overweight with healthy values of TC, TG, HDL-C, TC/HDL-C, glucose, insulin, HOMA, CRP, lower BP, and higher CRF compared to PRE).

### 3.2. POST versus 6-Month Follow-up Changes

A one-way repeated measures ANOVA found that between POST and 6 M follow-up in the SupExT group, AST (Δ = 2.4 U/L, 95% CI = 0.1, 4.7 U/L), GGT (Δ = 3.5 U/L, 95% CI = 0.1, 6.9 U/L), TC (Δ = 11.1 mg/dL, 95% CI = 4.4, 17.8 mg/dL), HDL-C (Δ = 2.8 mg/dL, 95% CI = 1.1, 4.4 mg/dL), LDL-C (Δ = 7.1 mg/dL, 95% CI = 1.2, 13.0 mg/dL), glucose (Δ = 5.0 mg/dL, 95% CI = 0.6, 9.4 mg/dL), HOMA-IR (Δ = 0.6, 95% CI = 0.1, 1.1), and HbA1C (Δ = 0.2%, 95% CI = 0.1, 0.3%) concentrations all significantly increased. There were no significant changes in all other biochemical markers. In the AC group, GGT (Δ = 8.2 U/L, 95% CI = 0.2, 16.2 U/L), TC (Δ = 8.9 mg/dL, 95% CI = 0.8, 17.0 mg/dL), and LDL-C (Δ = 9.2 mg/dL, 95% CI = 0.6, 17.7 mg/dL) were significantly raised from POST to 6 M follow-up. However, the two-way repeated measures ANOVA did not show an interaction of time by group for POST versus 6 M follow-up. Considering biochemical profile at 6 M follow-up, participants were again classified as MUO due to the following cardiometabolic abnormalities: obesity, higher values of TC, LDL-C, glucose, BP, and lower CRF compared to POST [14].

### 3.3. PRE versus 6-Month Follow-up Changes

Analysing the change from PRE to 6 M follow-up for the groups, a one-way repeated measures ANOVA found that, at 6 M follow-up, the SupExT group significantly reduced concentrations of CRP (Δ = −1.2 mg/L, 95% CI = −2.2, −0.2 mg/L), ALT (Δ = −6.3 U/L, 95% CI = −12.6, −0.1 U/L), GGT (Δ = −5.6 U/L, 95% CI = −11.1, −0.3 U/L), TG (Δ = −16.3 mg/dL, 95% CI = −26.4, −6.2 mg/dL), and TC/HDL-C (Δ = −0.3, 95% CI = −0.5, −0.1), and raised values in HDL-C (Δ = 2.8 mg/dL, 95% CI = 1.3, 4.4 mg/dL). No significant changes were found in any variable in the AC group. As changes occurred between PRE versus POST and POST versus 6 M follow up, no interaction of time by group was found for changes in PRE versus 6 M follow-up.

### 3.4. Medication Intake

Regarding medication intake, 89% of SupExT participants and 88% of AC participants were taking at least one medication at PRE. In SupExT, this percentage was significantly reduced in POST (83%, *p* < 0.05) and in 6 M follow-up (79%, *p* < 0.001). Although there was a reduction between POST and 6 M follow-up, the differences were not significant. In AC, 85.7% of participants were taking medication at POST, but the reduction was not significant from PRE (*p* > 0.05). There was no change from POST to 6 M follow-up (85.7%). From SupExT participants, 25.2% reduced the dose of their medication from PRE to POST, and 5.5% from POST to 6 M. Meanwhile, for the AC, 10.0% reduced the dose of their medication from PRE to POST, with no change from POST to 6 M follow-up. In particular, changes in SupExT medication intake were observed for statins (PRE 16.6%, POST 13.8%, 6 M follow-up 13.1%), Angiotensin-converting-enzyme inhibitors (ACEIs) (PRE 34.5%, POST 30.3%, 6 M follow-up 27.6%), and diuretics (PRE 42.1%, POST 37.9%, 6 M follow-up 36.6%). In AC, changes were only observed in ACEIs (PRE 49.0%, POST 42.9%, 6 M follow-up 42.9%). No significant change was observed in other medications. Although the separate analysis by group found more changes in SupExT than in the AC, the chi-square test revealed that there were no significant between-group differences in medication reduction. The percentage of smokers remained the same at PRE, POST, and 6 M follow-up in both groups.

## 4. Discussion

The current study demonstrated that a 16-week SupExT intervention with hypocaloric diet significantly improved cardiometabolic profiles from MUO to MHO. However, this was a transient stage as after 6 M of no supervision, participants’ biochemical profiles and CRF had returned to MUO.

There is now increasing evidence that obese individuals with a healthy lifestyle pattern could have a similar cardiovascular risk as healthy individuals with no obesity, since adherence to exercise and healthy diet together lead to beneficial changes in body composition and cardiometabolic profile [32]. The present study showed that the MUO profile in the EXERDIET-HTA group (i.e., obesity, HTN, fasting glucose > 100 mg/dL, LDL-C > 100 mg/dL, TC > 200 mg/dL, ALT > 30 U/L, systemic inflammation with CRP > 3 mg/L, and low CRF, 22.4 ± 5.4 mL·kg^−1^·min^−1^, Figure 2) was significantly worse than the HEALTHY non-obese group (Table 1). These results confirm that obesity and physical inactivity are the major contributing factors to dyslipidaemia manifested by elevated LDL-C and TG [33], inducing chronic inflammation and activation of the renin angiotensin system, and exacerbating, consequently, the sympathetic activation and BP [3].

The changes observed after conducting the 16-week exercise and diet intervention revealed that the significantly better body composition and CRF (results published in previous papers) [11,14] may be associated with a metabolic protective effect which improves insulin sensitivity, lipid panel, and BP (Table 2, Figure 2). Thus, the whole sample significantly decreased their CRP (Δ = −1.4 mg/L) concentrations (i.e., inflammatory marker associated with cardiovascular disease) which could be explained by multiple mechanisms, including a decrease in cytokine production by adipose tissue, skeletal muscles, endothelial and blood mononuclear cells, improved insulin sensitivity and endothelial function, and possibly an antioxidant effect [34]. Interestingly, after the 16-week intervention, only the SupExT group showed significant and beneficial changes in all variables, except HDL-C (Table 2). Thus, lower concentrations of hepatic enzymes ALT (Δ = −9.1 U/L) and GGT (Δ = −9.1 U/L) were shown, which in elevated concentrations are both highly related to liver and abdominal fat, and predictors of the MUO phenotype, the diagnosis of non-alcoholic fatty liver disease, type 2 diabetes, and subclinical atherosclerosis [35]. Previously, exercise and hypocaloric diet interventions have been shown to improve hepatic lipid composition via different pathways [36]. Thus, some studies suggest that exercise could modulate liver fat by directly altering hepatic lipid oxidation and lipogenesis [37], and that the improvement may be driven by adiponectin and insulin sensitivity [38]. As such, this may in part explain some of the beneficial changes seen in Table 2 after the 16-week intervention, including: lower concentrations of glucose (Δ = −6.8 mmol/L), insulin (Δ = −2.2 mU/L), HOMA-IR (Δ = −0.9 %), and HbA1C (Δ = −0.2 %). These markers have previously been shown to be some of the chronic exercise adaptations seen in skeletal muscle after the upregulation of muscle GLUT4 protein, increased enzyme capacities, and muscle capillarisation [39]. Similar to previous studies investigating the beneficial effects of SupExT [33,40], the current study found that lipid profile improved with lower concentrations of TC (Δ = −13.9 mmol/L), LDL-C (Δ = −9.4 mmol/L), TG (Δ = −20.9 mmol/L), and TC/HDL-C (Δ = −0.3) at POST. Conversely, in the current study, HDL-C concentrations (upper normal values at baseline) did not change after the exercise intervention. In other studies, which included just SupExT with no diet intervention, it was common to find elevated concentrations of HDL-C [15,33,41,42]. One possible explanation for this could be the incorporation of the DASH diet in the current study. The DASH diet has been previously revealed to lower HDL-C concentrations, along with TC, LDL-C [43,44], and TG [45]. The effect of the hypocaloric DASH diet in the intervention must be considered, since it has been indicated that it is also effective at changing the biochemical profile, i.e., lowering concentrations of CRP, AST, ALT, insulin, and HOMA-IR [45,46,47]. Further, the benefits produced by SupExT are notable in CRF and body composition data (published data for the same EXERDIET-HTA sample show greater benefits in SupExT compared to AC) [11,14]. Given the strong association of CRF with metabolic risk [48], there appears to be a strong rationale for including the CRF level in the prognosis of MHO to improve the stratification in individuals with obesity [5], empowering the fat-but-fit paradigm and SupExT [49]. 

Although the aforementioned results show the benefits that aerobic exercise could add to the hypocaloric DASH diet with respect to the biochemical profile (i.e., lower concentrations of glucose, insulin, HOMA-IR, LDL-C, TC, TG, ALT, GGT, CRP), in the present study, when the ANCOVA analysis was performed, no differences in the delta score (*p* > 0.05) were found between AC and SupExT for any variables. However, considering that AC had fewer participants (*n* = 43) than SupExT (*n* = 134), this lack of significance may have been due to power and thus, more research is needed to confirm or deny this.

Based on findings from the current study and that previously presented by the research team [11,14], the 16-week intervention was effective in changing the metabolic profile from MUO to MHO according to the following criteria (Figure 2) [7]: from obesity to overweight lowering body fat, with TC, ALT, glucose, and CRP moving from unhealthy to optimal cut-off values, reflecting the absence of metabolic abnormalities and lower levels of systemic inflammatory mediators, and CRF from “percentile under 50” to “percentile upper 50” for $\overset{\cdot }{V}$O_2peak_ classification [29]. Further, participants reduced their BP values (−5.4% reduction in MBP), and more than 7% of the participants stopped taking the medication and 25% reduced their doses under medical supervision [11,14].

Although there was a significant improvement in the metabolic profile after the 16-week intervention in the current study, there was subsequently a significant decline after 6 M when supervision was removed (Table 2, Figure 2). The MHO panel at POST was unable to be maintained after 6 M and participants regressed to MUO (Figure 2). The decrease in the lipidic, glycaemic, and haemodynamic profiles found after 6 M of no supervision in the present study is consistent with other studies which used different populations and with alternate types of exercise, duration of intervention, and non-supervised time [36,41,42,50]. These negative effects may be secondary to detraining-induced gains in body fat, favouring a more inflammatory status, and decreased CRF, as observed in participants. Previous studies have shown the pathophysiology of obesity related HTN [2]. Thus, an increase in the waist-to-hip ratio, parallel to a higher level of insulin, leptin, and the renin-angiotensin-aldosterone system seems to stimulate the sympathetic nervous system and concomitant increases in BP [2]. Further, it has already been established that a lower CRF, promoted by detraining, enhances the risk of suffering from metabolic syndrome and detrimental effects to the cardiovascular system, such as lack of regulation in BP, heart rate variability, myocardial oxygen demand, endothelial function, and systemic inflammation, in conjunction with inefficient fat storage [7]. Hence, it seems clear that the physical activity level differs between MHO and MUO in adults [1,5].

This suggests that supervision or alternative strategies of exercise provision are required given the need for MHO profile not to be a transient condition toward metabolic deterioration and consequently, a higher risk of developing cardiovascular disease [51,52]. Thus, in the present study, although the adherence with the 16-week intervention was very high in the SupExT, 6 M later, only 51% of all participants were engaged in physical activity ( >2 times per week) and implementing the recommendations (unpublished data from the EXERDIET-HTA study). However, it is interesting to note that HDL-C concentrations were better at 6 M follow-up than in PRE (Δ = 2.4 mmol/L) and POST (Δ = 2.5 mmol/L). This may be due to the effects of a non-supervised diet being a prominent factor in HDL-C change, as previously discussed with the DASH diet [43,44].

In order to interpret findings from the current study, it is essential to consider both the strengths and limitations. Since pre-intervention measurements, 23 individuals did not finish the intervention and 19 participants did not attend the 6 M follow-up measurements, making the sample smaller, which may have affected the between-group differences. Further, although medication intake and smoking status was noted, it is difficult to assess the possible influence this had on the results. Dietary intake was self-reported through questionnaires at PRE and POST, but it was not assessed at 6 M follow-up, precluding knowledge about the adherence to the DASH diet.

Future areas for research could determine whether exercise and diet supervision can maintain the achieved improvements in cardiometabolic health, by comparing 6 M with supervision and without supervision. Further, it would be interesting to understand the reasons for the deterioration of cardiometabolic health when the supervision is removed. Lastly, analysing the physical activity and diet during the unsupervised period may provide information about the time course of these changes. 

## 5. Conclusions

In conclusion, a 16-week SupExT and diet intervention was effective for improving cardiometabolic panel from MUO to MHO in a non-physically active population with obesity and HTN. However, this was a transient stage as after 6 M follow-up, participants returned to MUO. The findings of this study highlight the need for regular, systematic, and supervised diet and SupExT programs to avoid subsequent declines in cardiometabolic health in people who are physically inactive with overweight/obesity and HTN.

## Figures and Tables

**Figure 1 ijerph-17-02830-f001:**
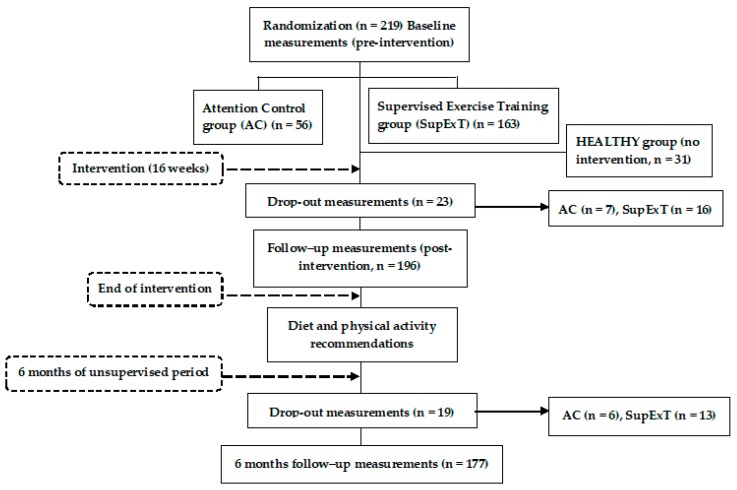
Flow diagram of the EXERDIET-HTA study from recruitment to the 6-month unsupervised period.

**Figure 2 ijerph-17-02830-f002:**
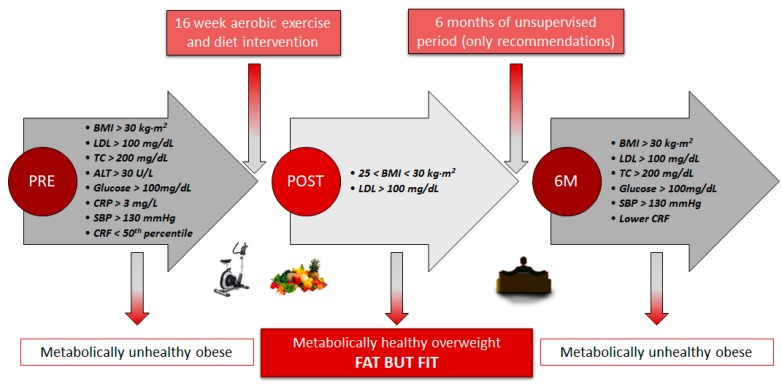
Participants’ profile from metabolically unhealthy obese to metabolically healthy overweight and back. BMI: Body mass index. LDL: Low-density lipoprotein cholesterol. TC: Total cholesterol. ALT: Alanine aminotransferase. CRP: C-reactive protein. SBP: Systolic blood pressure. CRF: Cardiorespiratory fitness.

**Table 1 ijerph-17-02830-t001:** Baseline results of the study population compared to a HEALTHY (normal body mass healthy population) group.

Variables	HEALTHY	EXERDIET-HTA	P PHEALTHY-EXERDIET-HTA
	(*n* = 31)	(*n* = 219)	
Age (years)	40.0 ± 9.0	53.3 ± 7.6	<0.001
Body mass (kg)	66.1 ± 10.5	92.4 ± 15.1	<0.001
BMI (kg/m^2^)	22.7 ± 2.2	32.4 ± 4.2	<0.001
Waist circumference (cm)	74.7 ± 8.0	103.5 ± 11.2	<0.001
FFM (%)	79.1 ± 6.2	65.0 ± 7.7	<0.001
SBP (mmHg)	114.0 ± 6.6	136.0 ± 11.8	<0.001
DBP (mmHg)	68.1 ± 7.2	78.0 ± 8.3	<0.001
MBP (mmHg)	83.4 ± 5.9	97.4 ± 8.5	<0.001
$\overset{\cdot }{V}$O_2peak_ (mL·kg^−1^·min^−1^)	48.0 ± 8.2	22.4 ± 5.4	<0.001
CRP (mg/L)	0.8 ± 0.7	4.1 ± 3.9	<0.001
AST (U/L)	21.9 ± 3.6	25.1 ± 9.4	0.001
ALT (U/L)	18.1 ± 5.9	33.3 ± 21.2	<0.001
GGT (U/L)	16.7 ± 8.3	40.1 ± 41.9	<0.001
TC (mg/dL)	180.9 ± 35.8	206.5 ± 38.3	0.001
HDL-C (mg/dL)	63.0 ± 10.8	47.2 ± 11.0	<0.001
LDL-C (mg/dL)	104.8 ± 30.3	132.0 ± 34.1	<0.001
TG (mg/dL)	70.6 ± 21.3	139.6 ± 78.9	<0.001
TC/HDL-C	2.9 ± 0.5	4.6 ± 1.5	<0.001
Glucose (mg/dL)	83.1 ± 9.3	101.9 ± 24.4	<0.001
Insulin (mU/L)	3.9 ± 1.3	12.3 ± 7.3	<0.001
HOMA-IR	0.8 ± 0.4	3.3 ± 2.4	<0.001
HbA1c (%)	5.5 ± 0.3	5.9 ± 0.8	0.019
**Medication Intake and Smoking Status**			
Statin (%)	0.0	14.2	0.024
Hypoglycaemic (%)	0.0	7.8	0.105
ACEI (%)	0.0	38.4	<0.001
ARB (%)	0.0	39.3	<0.001
Diuretic (%)	0.0	38.8	<0.001
CCB (%)	0.0	14.2	0.024
BB (%)	0.0	6.8	0.129
Antiplatelet (%)	0.0	3.7	0.275
Smokers (%)	15.6	11.0	0.533

BMI: Body mass index. FFM: Fat-free mass SBP: Systolic blood pressure. DBP: Diastolic blood pressure. MBP: Mean blood pressure. $\overset{\cdot }{V}$O_2peak_: Peak oxygen consumption. CRP: C-reactive protein. AST: Aspartate aminotransferase. ALT: Alanine transaminase. GGT: Gamma-glutamyl transpeptidase. TC: Total cholesterol. HDL-C: High-density lipoprotein cholesterol. LDL-C: Low-density lipoprotein cholesterol. TG: Triglycerides. HOMA-IR: HOmeostatic Model Assessment-Insulin resistance index. HbA1c: haemoglobin A1c. ACEI: Angiotensin-converting-enzyme inhibitors. ARB: Angiotensin II receptor blockers. CCB: Calcium channel blockers. BB: Beta-blockers.

**Table 2 ijerph-17-02830-t002:** Changes in metabolic panel at pre-intervention (PRE), immediately post-intervention (POST), and 6 months post supervised exercise cessation (6 M).

Variables	All	Effect	AC	ExT	Time × Group	*p* Groups PRE-POST	*p* Groups POST-6 M	*p* Groups PRE-6 M
	(*n* = 177)	Size (η_p_^2^)	(*n* = 43)	(*n* = 134)				
**CRP (mg/L)**								
PRE	4.0 ± 3.9 *	0.113	3.2 ± 3.2 *	4.1 ± 4.0 *	0.595	0.677	0.937	0.361
POST	2.6 ± 2.6		2.2 ± 2.1	2.7 ± 2.7				
6 M	2.9 ± 3.0 ^φ^		2.7 ± 2.5	3.0 ± 3.1 ^φ^				
**AST (U/L)**								
PRE	24.7 ± 8.6 *	0.060	25.2 ± 5.8	24.6 ± 9.2	0.465	0.684	0.380	0.657
POST	22.6 ± 8.5 ^$^		21.3 ± 5.0	22.9 ± 9.2 ^$^				
6 M	25.0 ± 11.3		23.9 ± 6.2	25.3 ± 12.3				
**ALT (U/L)**								
PRE	33.4 ± 20.8 *	0.112	29.4 ± 12.8	34.4 ± 22.3 *	0.529	0.090	0.103	0.923
POST	25.1 ± 15.4		24.2 ± 14.7	25.3 ± 15.7				
6 M	27.9 ± 20.4		27.3 ± 13.6	28.1 ± 21.9 ^φ^				
**GGT (U/L)**								
PRE	36.1 ± 24.1 *	0.155	36.2 ± 24.8	36.1 ± 24.1 *	0.230	0.914	0.152	0.459
POST	27.4 ± 18.4 ^$^		28.9 ± 27.6 ^$^	27.0 ± 15.5 ^$^				
6 M	31.8 ± 20.6		37.1 ± 33.5	30.4 ± 15.9 ^φ^				
**TC (mg/dL)**								
PRE	209.2 ± 36.3 *	0.103	207.3 ± 35.7	209.7 ± 36.6 *	0.281	0.102	0.330	0.195
POST	197.1 ± 35.4 ^$^		202.1 ± 35.9 ^$^	195.8 ± 35.2 ^$^				
6 M	207.8 ± 36.1		211.0 ± 39.7	206.9 ± 35.2				
**HDL-C (mg/dL)**								
PRE	48.6 ± 11.0	0.090	47.7 ± 8.0	48.8 ± 11.7	0.419	0.177	0.421	0.615
POST	48.5 ± 11.2 ^$^		47.1 ± 7.9	48.9 ± 12.0 ^$^				
6 M	51.0 ± 12.7 ^φ^		48.8 ± 8.6	51.6 ± 13.6 ^φ^				
**LDL-C (mg/dL)**								
PRE	135.2 ± 33.5 *	0.056	133.8 ± 35.4	135.6 ± 33.1 *	0.228	0.566	0.889	0.121
POST	127.3 ± 31.6 ^$^		131.2 ± 33.4 ^$^	126.2 ± 31.1 ^$^				
6 M	134.8 ± 31.5		140.4 ± 36.0	133.3 ± 30.1				
**TG (mg/dL)**								
PRE	125.2 ± 49.8 *	0.092	121.1 ± 38.2	126.3 ± 52.6 *	0.081	0.790	0.111	0.420
POST	108.2 ± 44.3		118.5 ± 45.4	105.4 ± 43.8				
6 M	109.8 ± 43.2 ^φ^		108.9 ± 41.8	110.0 ± 43.7 ^φ^				
**TC/HDL-C**								
PRE	4.6 ± 1.6 *	0.050	4.5 ± 1.1	4.5 ± 1.2 *	0.130	0.723	0.588	0.609
POST	4.3 ± 1.2		4.4 ± 1.2	4.2 ± 1.2				
6 M	4.4 ± 1.2 ^φ^		4.5 ± 1.1	4.2 ± 1.0 ^φ^				
**Glucose (mg/dL)**								
PRE	102.3 ± 25.7 *	0.043	96.9 ± 12.6	104.0 ± 28.3 *	0.392	0.098	0.137	0.765
POST	96.8 ± 22.5 ^$^		95.2 ± 11.4	97.2 ± 24.9 ^$^				
6 M	101.1 ± 29.7		97.1 ± 16.0	102.3 ± 32.7				
**Insulin (mU/L)**								
PRE	11.5 ± 6.1 *	0.077	10.6 ± 5.5	11.8 ± 6.3 *	0.492	0.576	0.679	0.827
POST	9.6 ± 6.0		9.5 ± 5.0	9.6 ± 6.3				
6 M	10.3 ± 5.9		9.3 ± 5.3	10.6 ± 6.1				
**HOMA-IR**								
PRE	3.1 ± 2.3 *	0.094	2.5 ± 1.3	3.3 ± 2.5 *	0.250	0.183	0.882	0.836
POST	2.3 ± 1.7 ^$^		2.2 ± 1.1	2.4 ± 1.8 ^$^				
6 M	2.8 ± 2.3		2.2 ± 1.5	3.0 ± 2.5				
**HbA1c (%)**								
PRE	5.9 ± 0.9 *	0.056	5.7 ± 0.3	6.0 ± 1.0 *	0.279	0.207	0.225	0.890
POST	5.7 ± 0.7 ^$^		5.6 ± 0.3	5.8 ± 0.7 ^$^				
6 M	5.9 ± 1.0		5.6 ± 0.4	5.9 ± 1.1				

AC: attention control group. ExT: Exercise training group. CRP: C-reactive protein. AST: Aspartate aminotransferase. ALT: Alanine transaminase. GGT: Gamma-glutamyl transpeptidase. TC: Total cholesterol. HDL-C: High density lipoprotein cholesterol. LDL-C: Low density lipoprotein cholesterol. TG: Triglycerides. HOMA-IR: HOmeostatic Model Assessment-Insulin resistance index. HbA1c: Haemoglobin A1c. * *p* < 0.005 intra-group PRE versus POST. ^$^
*p* < 0.005 intra-group POST versus 6 M. ^φ^
*p* < 0.005 intra-group PRE versus 6 M.

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
