# Peer review of "A Metabolically Healthy Profile Is a Transient Stage When Exercise and Diet Are Not Supervised: Long-Term Effects in the EXERDIET-HTA Study"

_ijerph, 2020, doi:10.3390/ijerph17082830_

Round 1
Reviewer 1 Report
The study was performed at a good methodological level using the right statistical criteria. The research topic is not new, but it confirms the basics of metabolic disorders in obesity. In general, the authors investigated blood biochemical parameters and anthropometric indicators. The work needs to be improved by completing the experimental part: an analysis of a wider range of indicators is needed to comment / explain the findings. At the moment, this is a statement of facts without any details or disclosure of mechanisms.
Authors need to make sure that self-plagiarism is not used in this manuscript.
The article uses old sources of literature, it is necessary to update and add more relevant information.
Author Response
REVIEWER 1
- The study was performed at a good methodological level using the right statistical criteria. The research topic is not new, but it confirms the basics of metabolic disorders in obesity. In general, the authors investigated blood biochemical parameters and anthropometric indicators.
- Thank you for your appreciation.
- The work needs to be improved by completing the experimental part: an analysis of a wider range of indicators is needed to comment / explain the findings. At the moment, this is a statement of facts without any details or disclosure of mechanisms.
- Thanks for your advice. A paragraph has been added in the discussion to complement the results related to detraining effects: Lines #341-350- These negative effects may be secondary to detraining-induced gains in body fat, favoring a more inflammatory status, and decreased CRF, as observed in participants. Previous studies have shown the pathophysiology of obesity-related hypertension. Thus, an increase in the waist-to-hip ratio, parallel to a higher level of insulin, leptin and the renin-angiotensin-aldosterone system seems to stimulate the sympathetic nervous system and concomitant increases in BP. Further, it has been already established that a lower CRF, promoted by detraining, enhances the risk of suffering from metabolic syndrome and detrimental effects to the cardiovascular system, such as lack of regulation in BP, heart rate variability, myocardial oxygen demand, endothelial function, and systemic inflammation, in conjunction with and inefficient fat storage. Hence, it seems clear that the physical activity level differs between MHO and MUO in adults.
- Authors need to make sure that self-plagiarism is not used in this manuscript.
- We have reviewed the manuscript and we have referenced all of our previous papers. We are presenting new data, and we are using previous published data to better understand the results and discussion.
- The article uses old sources of literature, it is necessary to update and add more relevant information.
- Thank you for your comment. We have reviewed all the references and most of them are recent studies or statements (2012-2019). Those older ones are associated to specific topics, which could be considered as seminal papers (e., those that explain well-known theoretical frameworks and concepts), such as DASH diet (2001, 2010), detraining (2006), IPAQ questionnaire (2003), HOMA assessment (1985), Adult Treatment Panel III (2002), Diabetes statement (2006), A1C for diagnosis of diabetes (2009), insulin resistance quantification (2001), C-reactive protein (2005, 2011), fatty liver disease (2010)
Reviewer 2 Report
In the future research, it would be interesting to compare 6M with supervision and without supervision.
Author Response
REVIEWER 2
- In the future research, it would be interesting to compare 6M with supervision and without supervision.
- Thank you for your observation, we will take it into account. We added a sentence about it in future areas for research: Lines #371-372- Future areas for research could determine whether exercise and diet supervision can maintain the achieved improvements in cardiometabolic health, by comparing 6M with supervision and without supervision.
Reviewer 3 Report
-The paper is very great for readers. However, I suggest to add in the abstract more number in results section. For eg. p value, delta values pre-post intervention.
-The lines 260-263 must be removed.
-Figure 2 is unecessary. Please, remove it.
-The data must be adjusted by medication use. The authors included as a limitation the medication use. But did not explore these informations. The ANCOVA adjusted by medication use is need.
-16-wk is not long-term. Please, revise!
Author Response
REVIEWER 3
- The paper is very great for readers. However, I suggest to add in the abstract more number in results section. For eg. p value, delta values pre-post intervention.
- Thank you for all your comments. We have reviewed the abstract and some requested values have been added. “From PRE to POST, MUO participants became MHO with improved (P<0.05) total cholesterol (TC, ∆=-12.1 mg/dL), alanine aminotransferase ∆=-8.3 U/L), glucose (∆=-5.5 mg/dL), C-reactive protein (∆=-1.4 mg/dL), systolic blood pressure (SBP) and CRF compared to unhealthy optimal cut-off values.”
- The lines 260-263 must be removed.
- Amended
- Figure 2 is unnecessary. Please, remove it.
- Agree that the values in figure 2 are repeated in the tables. But we think that the graphical presentation of these results is a great visual communication to use in oral presentations and social media. In this sense, it would also be a good opportunity for the journal to get more impact. Anyway, if you and the editor of the journal think that it is better to delete Figure 2, please remove it.
- The data must be adjusted by medication use. The authors included as a limitation the medication use. But did not explore these informations. The ANCOVA adjusted by medication use is needed.
- Thank you for your appreciation, as you suggested, the ANCOVA analysis is now adjusted by medication use (line #172, Table 2). No relevant changes were observed due to this new adjustment in the results.
- -16-wk is not long-term. Please, revise!
- When we use long-term concept is not related to supervised exercise period (intervention), but to the detraining period, which 6 months-periods is considered long-term. Anyway, we have reviewed all the manuscript and we think that there are no errors in the terminology.
Reviewer 4 Report
The study by Corres et al summarized the importantfindings which highlighted the need for regular, systematic and supervised diet and exercise programs to avoid subsequent declines in cardiometabolic health. The manuscript is well written, has important clinical message, and should be of great interest to the readers. However, below are some concerns regarding the study-
Technical comments to author-
- If possible, author should mention the inclusion and exclusion criteria regarding the smokers and alcoholics in the material method section of the study. Also, about hypocholesterolemic drugs if taken by patients.
- Author did not mention about Glucose Tolerance Test in these subjects which would have provided an idea about insulin resistance and Haemoglobin A1c levels in these subjects. If possible, author can add this data which will provide more scientific soundness to the study.
Author Response
REVIEWER 4
The study by Corres et al summarized the important findings, which highlighted the need for regular, systematic and supervised diet and exercise programs to avoid subsequent declines in cardiometabolic health. The manuscript is well written, has important clinical message, and should be of great interest to the readers. However, below are some concerns regarding the study.
Technical comments to the author-
- If possible, author should mention the inclusion and exclusion criteria regarding the smokers and alcoholics in the material method section of the study. Also, about hypocholesterolemic drugs, if taken by patients.
- Thank you for your appreciation. Smoking was not an exclusion criterion, and in Table 1 is described that 11.0% of the participants of the EXERDIET-HTA study were smokers. Similarly, hypocholesterolemic drugs were allowed for participants of the study. In this sense, in Table 1 the medication intake of participants can be observed, with 14.2% of them taking statins. About the medication use, another reviewer suggested adjusting the ANCOVA analysis also with medication intake, and as you can see in this new version of the manuscript, the requested analysis has been already performed. Concerning alcoholism, this criterion was not included in the study protocol as an exclusion criterion when the study was devised. However, dietary questionnaires were used, and we can confirm that there was no participant with alcoholism. The main inclusion criteria have been presented in the methods. For the exclusion criteria, the protocol study paper [1] (line #99) has been referred not to make the method section too long.
- Author did not mention about Glucose Tolerance Test in these subjects which would have provided an idea about insulin resistance and Haemoglobin A1c levels in these subjects. If possible, author can add this data which will provide more scientific soundness to the study.
- Regrettably, the Glucose Tolerance Test was not performed in this study, so we cannot add this data. However, we performed the fasting glucose test and HbA1c test. The insulin resistance index (HOMA-IR) was determined by [fasting serum insulin (µU/mL) x fasting plasma glucose (mg/dL)/405].
Other changes: We perceived a mistake in an abbreviation (Ext instead of SupExt) in Figure 1, so the figure is replaced for the correct one.
REFERENCE
- Maldonado-Martín, S.; Gorostegi-Anduaga, I.; Aispuru, G.; Illera-Villas, M.; Jurio-Iriarte, B.; Francisco-Terreros, S.; Pérez-Asenjo, J. Effects of different aerobic exercise programs with nutritional intervention in primary hypertensive and overweight/obese adults: EXERDIET-HTA controlled trial. J. Clin. Trials 2016, 6, 1-10. 10.4172/2167-0870.1000252 [doi].
Round 2
Reviewer 1 Report
The authors made all the necessary changes. I recommend to accept for publication
Reviewer 3 Report
-Figure 2: To remove.
-Abstract: To add the SD values.